# A Real-Time Dual-Microphone Speech Enhancement Algorithm Assisted by Bone Conduction Sensor

**DOI:** 10.3390/s20185050

**Published:** 2020-09-05

**Authors:** Yi Zhou, Yufan Chen, Yongbao Ma, Hongqing Liu

**Affiliations:** 1School of Communication and Information Engineering, Chongqing University of Posts and Telecommunications, Chongqing 400065, China; zhouy@cqupt.edu.cn (Y.Z.); S180101136@stu.cqupt.edu.cn (Y.C.); 2Suresense Technology, Chongqing 400065, China; mayongbao@suresense.tech

**Keywords:** array signal processing, bone conduction, beamforming, speech enhancement, deep learning, real time

## Abstract

The quality and intelligibility of the speech are usually impaired by the interference of background noise when using internet voice calls. To solve this problem in the context of wearable smart devices, this paper introduces a dual-microphone, bone-conduction (BC) sensor assisted beamformer and a simple recurrent unit (SRU)-based neural network postfilter for real-time speech enhancement. Assisted by the BC sensor, which is insensitive to the environmental noise compared to the regular air-conduction (AC) microphone, the accurate voice activity detection (VAD) can be obtained from the BC signal and incorporated into the adaptive noise canceller (ANC) and adaptive block matrix (ABM). The SRU-based postfilter consists of a recurrent neural network with a small number of parameters, which improves the computational efficiency. The sub-band signal processing is designed to compress the input features of the neural network, and the scale-invariant signal-to-distortion ratio (SI-SDR) is developed as the loss function to minimize the distortion of the desired speech signal. Experimental results demonstrate that the proposed real-time speech enhancement system provides significant speech sound quality and intelligibility improvements for all noise types and levels when compared with the AC-only beamformer with a postfiltering algorithm.

## 1. Introduction

In recent years, the signal transmission bandwidth and network technology have been significantly improved, and therefore, communication system can real-time transmit speech signals with a higher sampling rate and deeper sampling bit depth. In speech communication systems, the sound field noise at the mobile communication terminal is the dominant aspect that degrades the communication quality [1]. Speech enhancement is a technology for improving the sound quality and intelligibility of speech signals in the acoustic front end. It uses various speech signal processing approaches in stationary and non-stationary noises, reverberant, point source interference, and diffuse field interference scenarios. Speech enhancement technology can be broadly divided into two categories according to the number of microphones used, i.e., single-channel and multi-channel [2]. 

Single-channel speech enhancement usually assumes that the noise is additive, and the statistical characteristics of the noise change much more smoothly compared to the speech. The noise is estimated by statistics or minima-controlled methods, and finally the gain function is calculated to complete the noise reduction, for example, spectral subtraction [3], Weiner filter [4], and optimally modified log-spectral amplitude (OMLSA) estimator [5], to name a few. Single-channel speech enhancement algorithms are difficult to effectively remove non-stationary noise and usually introduce additional musical noise [1]. In recent studies, the single-channel speech enhancement technology based on a deep neural network (DNN) has been proven to be effective to suppress transient non-stationary noise [6]. However, DNN-based methods often have a large number of parameters and high computational complexity, making it difficult to be deployed in real-time systems with low power consumption requirements. Recently, Valin [7] proposed a low computational complexity method, recurrent neural network (RNNoise), which combines deep learning and digital signal processing and can process 48 kHz sampling rate audio on a low-power central processing unit (CPU) in real time. However, this lightweight network lacks a powerful for front-end processing method like microphone array to improve the signal-to-noise ratio (SNR) to ensure acceptable performance.

The fundamental difference between single-channel and multi-channel speech enhancement algorithms is that the multi-channel method utilizes the spatial information with multiple microphone sensors. The interferences from undesired directions will be attenuated by the spatial filter composed of the microphone array. Adaptive beamforming is a widely used microphone array speech enhancement algorithm [8], which updates the weight coefficients according to the relationship between the interference and the desired signal in a frame-by-frame manner. Many studies have been conducted to improve the speech quality and intelligibility using microphone arrays and beamforming technology. The minimum variance distortionless response (MVDR) beamformer minimizes the variance of the noise while maintaining the desired signal distortion free. Habets [9] implemented an MVDR beamformer in room acoustics for speech denoising and dereverberation. The least constraint minimum variance (LCMV) [10] method was developed with additional linear constraints, and a variant of LCMV, the generalized sidelobe canceller (GSC), was proposed by Griffiths and Jim [11]. The GSC is composed of three blocks: a fixed beamformer (FBF), a signal-blocking matrix (BM), and an adaptive noise canceller (ANC). FBF is a spatial filter with fixed weight coefficients which retains the desired signal and suppresses the undesired signals. The BM blocks the desired speech components in reference noise signals and ANC eliminates the noise components leaking in the sidelobes of FBF. The work of this paper extends the GSC beamforming structure.

The above discussed methods are based on air-conduction (AC) microphone arrays and showed limited performance in low SNR and highly diffuse noise environments [12]. Recently, the internet of things (IOT) network and devices have developed rapidly. Those technologies have been applied in industry and consumer electronics [13,14,15,16]. As a micro-electro-mechanical system (MEMS) sensor, the bone-conduction (BC) sensor collects vibration signals transmitted from the vocal cords to the bones. Unlike the AC microphone, the BC sensor is relatively insensitive to the ambient acoustic noises and interferences that are conducted by air [17]. However, there is a non-trivial issue that the high-frequency components of the BC signal are seriously attenuated, and the low-frequency components are not exactly the same as those of the AC signal, due to the transmission loss and the sensitivity of the sensor. This problem is particularly prominent for low-cost mass production sensors. Therefore, an additional enhancement of the BC signal is necessary. Several algorithms have been proposed to use the BC signal to aid speech enhancement [17]. In [18], a method was proposed to find a non-linear mapping mechanism between BC and AC signals using a DNN-based autoencoder to enhance the speech signal. The offline training process requires a high sensor quality with less noise and higher sensitivity. In [19], an approach using a bandwidth extension of the BC signal via an adaptive filter for speech enhancement was proposed. Shin [20] proposed an approach using AC and BC signals to estimate a prior SNR. The BC sensor was deployed to assist the noise estimation in [21]. However, using BC signal characteristics to assist microphone array beamforming based on a spatial adaptive filter is rare. It is known that the microphone array beamformer yields a significant improvement in SNR and speech quality [8]. However, the noise suppression is insufficient when the spatial discrimination between noise and desired speech signal is low or the noise field is diffuse [22]. To reduce this kind of residual noise, an additional postfiltering is usually required. The research on the postfiltering algorithm of the microphone array can be divided into two groups. The first is to process the output of the adaptive beamformer with a single channel speech enhancement algorithm [1,23,24]. The second multichannel postfilters utilize the spatial information in the GSC structure and the relationship between the beam and the interference to distinguish between speech and non-stationary noise and thereby obtains a better postfiltering noise suppression capability [25,26,27]. In the non-stationary noise sound field, when the noise is a non-point source or the noise source is in the same direction with the desired signal, the array postfiltering algorithm has difficulty obtaining benefit from the spatial information provided by the beamformer. To combine a real-time simple recurrent unit (SRU)-based neural network noise reduction algorithm with a small number of parameters, this paper treats the array postfiltering as a single-channel noise reduction task.

In this paper, a GSC-based and BC signal-assisted dual-microphone adaptive beamforming algorithm with an SRU-based postfilter was proposed. An improved voice activity detection (VAD) was extracted from the BC, which effectively assists the weight coefficient update of the ANC and the adaptive block matrix (ABM). We designed a filter that was named a compensation filter (CF) in this paper to learn the delay and amplitude difference between the BC and the AC signals. The BC signal is filtered to approximate the clean AC signal and is fused with the output of the ANC in an appropriate ratio to further suppress the noise. To combat the wind noise, a dual-microphone wind noise detector is deployed and the wind noise portion will be replaced by the output of the CF. With the assistance of the BC signal, the noise suppression capability of the dual-microphone GSC system under low SNR or harsh conditions can be improved effectively. Finally, a lightweight real-time single-channel noise reduction system based on SRU was used as a postfilter to eliminate the residual noise in the microphone array output.

The rest of this paper was organized as follows: the proposed dual-microphone array assisted by a BC signal approach was introduced in Section 2. The integration of a BC-assisted microphone array and SRU-based postfilter is presented in Section 3. Experimental details are described in Section 4. Results and discussion are shown in Section 4. Finally, a conclusion of this paper is given in Section 5.

## 2. Dual-Microphone Array Assisted by BC Signal

### 2.1. Signal Model

In this work, we considered a frame-by-frame processing of an AC dual-microphone array and BC signals in the frequency domain. The dual-microphone array can be treated as a uniform linear array (ULA), depicted in Figure 1. It is important to estimate the position of the sound source for the microphone array signal processing. The direction of arrival (DOA) was used to obtain the steering vector. There are many methods to locate the direction of the source indoors [28,29,30]. Since the algorithm application scenario proposed in this paper is a wearable device, we assume that the interference source was in the broadside direction and that the desired source was fixed in the end fire direction, along the axis of the array. The observed signal in the time domain at the *m*-th sensor (m=1,2) is:(1)$$y_{m}(t)=x(t)*s_{m}(t)+v_{m}(t)$$
where ∗ denotes convolution. x(t) is the desired speech signal, $v_{m}(t)$ contains the directional and non-directional noises, and $s_{m}(t)$ is the system function that represents the acoustic wave propagation delay and the room impulse response (RIR). 

Applying short-time Fourier transform (STFT) to (1) yields its complex spectral time–frequency domain expression as
(2)$$\begin{array}{cc} y(k,l) & =X(k,l)s(k,l)+v(k,l)\\ & =X_{1}(k,l)d(k,l)+v(k,l) \end{array}$$
where: (3)$$d(k,l)={\left[1,\frac{S_{2}(k,l)}{S_{1}(k,l)}\right]}^{T}$$
(4)$$\mathrm{and}\;v(k,l)={\left[V_{1}(k,l),V_{2}(k,l)\right]}^{T}$$

l and k are the time-frame and frequency-bin indices, respectively, and $X_{1}(k,l)$ is the desired AC signal collected by the reference microphone in the frequency domain. Since the desired signal impinges on the ULA from the end fire direction, considering the delay-only model, the steering vector d(k,l) is:(5)$$d(k)={\left[1,e^{-j\frac{2\pi }{k}\tau }\right]}^{T}$$
where τ=d/c and d is the distance between two microphones, c is the speed of sound.

### 2.2. Reliable VAD from BC Signal

#### 2.2.1. BC Signal

Unlike the AC microphone, the BC sensor is relatively insensitive to the ambient acoustic noise and interferences which are conducted by air. Due to the transmission loss and the sensitivity of the sensor, the high-frequency components of the BC signal are seriously attenuated, and the low-frequency components are not exactly the same as those of the AC signal. This problem is particularly prominent for low-cost mass production sensors. In Figure 2, it can be observed that the BC signal suffers from a significant attenuation around 1.5 kHz and above, and the sensor thermal noise is also present. 

#### 2.2.2. BC VAD Estimator

In order to obtain an accurate and robust VAD from the noisy BC signal, we used a simple noise estimator. The BC speech that is corrupted by the uncorrelated additive thermal noise is given by
(6)$$Y_{\mathrm{BC}}(k,l)=S(k,l)+N(k,l)$$

To derive the VAD estimator, it is assumed that the thermal noise and the BC speech spectral coefficients have a complex Gaussian distribution [31]. Two hypotheses for VAD are:$$H_{0}:\;\mathrm{speech}\;\mathrm{absent}:\;Y=N\;H_{1}:\;\mathrm{speech}\;\mathrm{present}:\;Y=N+S$$

The probability density functions (PDF) of the two hypotheses $H_{0}$ and $H_{1}$ are written as
(7)$$p(Y|H_{0})=\prod_{k=0}^{L-1}\frac{1}{\pi \lambda_{N}(k)}\exp(-\frac{|Y(k){|}^{2}}{\lambda_{N}(k)}),$$
(8)$$\mathrm{and}\;p(Y|H_{1})=\prod_{k=0}^{L-1}\frac{1}{\pi (\lambda_{N}(k)+\lambda_{s}(k))}\exp(-\frac{|Y(k){|}^{2}}{\lambda_{N}(k)+\lambda_{s}(k)})$$
where L, $\lambda_{N}(k)$, and $\lambda_{s}(k)$, respectively, denote the coefficients’ dimension of the discrete Fourier transform (DFT), the variance of N(k,l) and the variance of S(k,l). The likelihood ratio is given by
(9)$$\Lambda =\frac{p(Y|H_{0})}{p(Y|H_{1})}$$

For mathematical convenience, the logarithm operation is applied which yields:(10)$$\begin{array}{cc} \log\Lambda & =\log\frac{p(Y|H_{0})}{p(Y|H_{1})}=\sum_{k=0}^{L-1}\left(\log(\frac{\left(\lambda_{N}(k)+\lambda_{s}(k))\exp(-\frac{|Y(k){|}^{2}}{\lambda_{N}(k)}\right)}{\lambda_{N}(k)\exp(-\frac{|Y(k){|}^{2}}{\lambda_{N}(k)+\lambda_{s}(k)})}\right)\\ & =\sum_{k=0}^{L-1}\log(\frac{1}{1+\xi (k)}\exp(\frac{\gamma (k)\xi (k)}{1+\xi (k)})) \end{array}$$
where $\xi (k)=\lambda_{s}(k)/\lambda_{N}(k)$ and $\gamma (k)=|Y(k){|}^{2}/\lambda_{N}(k)$ are called a priori and a posteriori SNRs. For a real-time implementation, γ(k) can be estimated as $\overset{⌢}{\gamma }(k)=|Y(k){|}^{2}/{\overset{⌢}{\lambda }}_{N}(k)$ and the estimate of ξ(k) can be treated as $\overset{⌢}{\xi }(k)=\max(0,{\overset{⌢}{\xi }}_{\mathrm{ml}}(k))=\max(0,\overset{⌢}{\gamma }-1)$ by a limited maximum likelihood (ML) [32]. Therefore, the frame VAD decision rule is given by
(11)$$f_{\mathrm{VAD}}(l)=\{\begin{array}{ccc} 1, & \; & \log\Lambda \geq \eta ,\\ 0, & \; & \log\Lambda <\eta . \end{array}$$

To obtain the noise variance ${\overset{⌢}{\lambda }}_{N}(k)$, it can be estimated by a simple recursive function as
(12)$${\overset{⌢}{\lambda }}_{N}(k,l)=\{\begin{array}{ccc} \alpha_{0}{\overset{⌢}{\lambda }}_{N}(k,l-1)+(1-\alpha_{0})|Y(k,l){|}^{2}, & \; & f_{\mathrm{VAD}}(l)=1,\\ {\overset{⌢}{\lambda }}_{N}(k,l-1), & \; & f_{\mathrm{VAD}}(l)=0. \end{array}$$

### 2.3. BC VAD-Assisted Adaptive Beamforming

#### 2.3.1. Robust Generalized Sidelobe Canceller

The robust GSC structure [33] is shown in Figure 3. The dual-microphone inputs are received by the fixed beamforming (FBF) that directs the beam towards the end fire desired signal direction, i.e., at 0°. $Y_{\mathrm{FBF}}(k,l)$ is used as the reference signal of the adaptive block matrix (ABM) to eliminate the desired signal components in BM. In order to suppress only those desired signals and maximize the noise reduction, the adaptive filter coefficients were constrained within DOA-based boundaries. Then, the output of ABM $U_{\mathrm{ABM}}(k,l)$ was passed to the adaptive noise canceller (ANC) as the reference signal to suppress the components that are correlated to the interference signal in the FBF output $Y_{\mathrm{FBF}}(k,l)$.

#### 2.3.2. Adaptive Block Matrix Assisted by BC Signal

The ABM utilizes an adaptive algorithm to suppress the speech portion of the reference microphone. Its output is
(13)$$U(k,l)\;=X_{1}(k,l)-G_{\mathrm{ABM}}^{H}(k,l)Y_{\mathrm{FBF}}(k,l)$$

Unlike the method in [33], where the weight coefficients are constrained by DOA boundaries, the reliable VAD obtained from the BC signal is used to control the weight coefficients update of the normalized least mean square (NLMS) adaptive filter of ABM, shown in Figure 4, which yields:(14)$$G_{\mathrm{ABM}}(k,l+1)=G_{\mathrm{ABM}}(k,l)+f_{\mathrm{VAD}}(l)\mu_{0}\frac{Y_{\mathrm{FBF}}(k,l)U^{\prime }(k,l)}{{\bar{p}}_{\mathrm{FBF}}(k,l)}$$
where $\mu_{0}=0.3$ is a step size constant when the speech is present. $U^{\prime }(k,l)$ represents the conjugate of U(k,l) and ${\bar{p}}_{\mathrm{FBF}}(k,l)$ is the smoothed power of $Y_{\mathrm{FBF}}(k,l)$ by a first order recursive smoothing as
(15)$${\bar{p}}_{\mathrm{FBF}}(k,l)=\alpha_{0}{\bar{p}}_{\mathrm{FBF}}(k,l-1)+(1-\alpha_{0})|Y_{\mathrm{FBF}}(k,l-1){|}^{2}$$

The adaptive filter update controller based on BC VAD does not need to estimate the DOA, and in the harsh sound field environment, it is also difficult to effectively estimate the accurate DOA.

#### 2.3.3. Adaptive Noise Canceler Assisted by BC signal

The ANC adaptively updates the filter $G_{\mathrm{ANC}}(k,l)$ with the minimum output power criterion and uses the ABM output U(k,l) as a reference signal to suppress the residual noise in the output of FBF, that is: (16)$$Y_{\mathrm{GSC}}(k,l)\;=Y_{\mathrm{FBF}}(k,l)-G_{\mathrm{ANC}}^{H}(k,l)U(k,l)$$

This adaptive problem can be solved by the following NLMS algorithm:(17)$$G_{\mathrm{ANC}}(k,l+1)=G_{\mathrm{ANC}}(k,l)+\mu_{\mathrm{ANC}}(k,l)\frac{U(k,l)Y_{\mathrm{GSC}}^{\prime }(k,l)}{{\bar{p}}_{U}(k,l)}$$

Similar to (15), ${\bar{p}}_{U}(k,l)$ is:(18)$${\bar{p}}_{U}(k,l)=\alpha_{0}{\bar{p}}_{U}(k,l-1)+(1-\alpha_{0})|U(k,l-1){|}^{2}$$
where $\mu_{\mathrm{ctrl}}(k,l)$ controls the update of the step size. Considering the signal to interference ratio (SIR) and the VAD of BC signal, $\mu_{\mathrm{ctrl}}(k,l)$ can be updated as
(19)$$\mu_{\mathrm{ctrl}}(k,l)=\frac{\mu_{0}}{SIR(k,l)\cdot f_{\mathrm{VAD}}(l)+1}$$
where: (20)$$SIR(k,l)=\frac{|Y_{\mathrm{FBF}}(k,l){|}^{2}}{|U(k,l){|}^{2}}.$$

The update mechanism is as follows. When $f_{\mathrm{VAD}}(l)$ is equal to 0, the algorithm tends to update with a larger step size, whereas when $f_{\mathrm{VAD}}(l)$ is equal to 1, the step size update depends on the value of SIR(k,l).

### 2.4. Robust Compensation Filter for Low Frequencies 

After the processing of BC-assisted GSC, an algorithm is developed by using low-frequency BC speech components to suppress the low-frequency noise in the final output. Another NLMS adaptive filter is employed, given by
(21)$$N(k,l)=Y_{\mathrm{GSC}}(k,l)-G_{\mathrm{CF}}(k,l)Y_{\mathrm{BC}}(k,l)$$
where $Y_{\mathrm{GSC}}(k,l)$ is the output of GSC and $G_{\mathrm{CF}}(k,l)$ represents the CF weight coefficient which can be updated by NLMS as
(22)$$G_{\mathrm{CF}}(k,l+1)=G_{\mathrm{CF}}(k,l)+\mu_{\mathrm{CF}}(k,l)\frac{Y_{\mathrm{BC}}(k,l)N^{\prime }(k,l)}{p_{u}(k,l)},$$
(23)$$\mathrm{and}\;{\bar{p}}_{u}(k,l)=\alpha_{1}{\bar{p}}_{u}(k,l-1)+(1-\alpha_{1})|Y_{\mathrm{BC}}(k,l){|}^{2}$$
where $\mu_{\mathrm{CF}}(k,l)$ is used to control the step size update of the CF, and its solution is: (24)$$\mu_{\mathrm{CF}}=\mu_{1}f_{\mathrm{VAD}}(l)$$
where $\mu_{1}$ is a constant controlling the overall step size of NLMS. This solution combines the BC and VAD, so that CF will be updated when speech is present and not be updated when speech is absent. Therefore, the output of CF is:(25)$$Y_{\mathrm{CF}}(k,l)=G_{\mathrm{CF}}(k,l)Y_{\mathrm{BC}}(k,l)$$

Based on the above derivations, an optimizer is designed to fuse the low-frequency part of $Y_{\mathrm{CF}}(k,l)$ with the GSC output so as to provide a reasonable compromise between the noise suppression and voice quality, given by
(26)$$Y_{\mathrm{LF}}(k,l)=p_{1}(k,l)Y_{\mathrm{GSC}}(k,l)+(1-p_{1}(k,l))Y_{\mathrm{CF}}(k,l),$$
(27)$$\mathrm{and}\;p_{1}(k,l)=tanh(SIR(k,l))=\frac{e^{SIR(k,l)}-e^{-SIR(k,l)}}{e^{SIR(k,l)}+e^{-SIR(k,l)}}$$
where $p_{1}(k,l)$ is a value that uses the tanh(x) function to map the SIR to [0.5, 1]. It increases (decreases) the proportion of $Y_{\mathrm{CF}}(k,l)$ when the SNR is low (high). This ensures the sound quality and intelligibility at high a SNR with good noise suppression capability. Considering the cut-off frequency bin of the BC sensor $k_{\mathrm{cutoff}}$, the output of the system can be written as
(28)$$Y_{\mathrm{Fused}}(k,l)=\{\begin{array}{cc} Y_{\mathrm{GSC}}(k,l), & k\geq k_{\mathrm{cutoff}},\\ \min(Y_{\mathrm{LF}}(k,l),Y_{\mathrm{GSC}}(k,l)), & k<k_{\mathrm{cutoff}}. \end{array}$$
where $\min(Y_{\mathrm{LF}}(k,l),Y_{\mathrm{GSC}}(k,l))$ protects the output signal from the thermal noise of the BC sensor when the output SNR of the GSC is high.

### 2.5. Wind Noise Suppression

In order to suppress the wind noise, we used the output of (21) $Y_{\mathrm{LF}}(k,l)$ to compensate the low-frequency speech components corrupted by wind noise. The processed BC and clean AC signal are very similar in the frequency spectrum, but simply replacing all the low-frequency parts of the AC signal by BC signals will impair the intelligibility of the GSC output at high SNR. To alleviate this problem, a wind noise detector is deployed here, and only the speech components corrupted by wind noise will be replaced by BC signals.

A wind noise detector based on the ratio of the cross power spectral density (PSD) $\Phi_{x_{1}x_{2}}(k,l)$ and the square root of the product of two auto-PSD $\Phi_{x_{1}x_{1}}(k,l)$ and $\Phi_{x_{2}x_{2}}(k,l)$ is utilized [34], which is given by
(29)$$\Gamma (k,l)=\frac{\Phi_{x_{1}x_{2}}(k,l)}{\sqrt{\Phi_{x_{1}x_{1}}(k,l)\Phi_{x_{2}x_{2}}(k,l)}}$$
where: (30)$$\Phi_{x_{1}x_{2}}(k,l)=\alpha_{\mathrm{sm}}\Phi_{x_{1}x_{2}}(k,l-1)+(1-\alpha_{\mathrm{sm}})Y_{1}(k,l){Y_{2}}^{\prime }(k,l),$$
(31)$$\Phi_{x_{1}x_{1}}(k,l)=\alpha_{\mathrm{sm}}\Phi_{x_{1}x_{1}}(k,l-1)+(1-\alpha_{\mathrm{sm}})Y_{1}(k,l){Y_{1}}^{\prime }(k,l),$$
(32)$$\Phi_{x_{2}x_{2}}(k,l)=\alpha_{\mathrm{sm}}\Phi_{x_{2}x_{2}}(k,l-1)+(1-\alpha_{\mathrm{sm}})Y_{2}(k,l){Y_{2}}^{\prime }(k,l)$$
where $\alpha_{\mathrm{sm}}=0.3$ is a factor for smoothing the PSD. The wind noise is directly produced by turbulences on a boundary layer close to the microphones. In [35], it is shown that the wind noise between the different microphones is uncorrelated. Therefore, a magnitude squared coherence (MSC) coefficient Γ(k,l) was used to describe the correlation of the two microphone signals. With MSC, an indicator of the existence of wind noise is obtained by comparing the ratio with a fixed threshold as follows:(33)$$r(k,l)=\{\begin{array}{cc} 0, & |\Gamma (k,l)|\geq \sigma_{1},\\ 1. & |\Gamma (k,l)|<\sigma_{1}. \end{array}$$
where $\sigma_{1}=0.35$ is a constant threshold. Finally, the output now is:(34)$$Y_{\mathrm{output}}(k,l)=r(k,l)Y_{\mathrm{GSC}}(k,l)+(1-r(k,l))Y_{\mathrm{LF}}(k,l)$$

## 3. Recurrent Neural Network-Based Real-Time Postfiltering with a Small Number of Parameters

### 3.1. Feature Compression

For training a neural network, it is important to find appropriate features, which affect the training efficiency and deduction performance of neural networks [36]. However, in many approaches summarized by the overview [6], neural networks are used to directly estimate the frequency bin ideal ratio mask (IRM) or magnitude and require a million weights to process 16 kHz speech. Obviously, these types of networks are difficult to deploy in low-power real-time systems. Inspired by the sub-band signal processing method [37], we divide the frequency bin spectrum into 40-dimensional band amplitudes of the Mel-scale [38] as the input features of the neural network, which is shown in Figure 5. The transfer function of the *m*-th filter is: (35)$$H_{m}(k)=\{\begin{array}{cc} 0, & k<f(m-1),\\ \frac{k-f(m-1)}{f(m)-f(m-1)}, & f(m-1)\leq k\leq f(m)\\ \frac{f(m+1)-k}{f(m+1)-f(m)}, & f(m)<k\leq f(m+1)\\ 0, & k>f(m+1) \end{array}$$
where:(36)$$f(m)=(\frac{N}{f_{s}})F_{\mathrm{mel}}^{-1}(F_{\mathrm{mel}}(f_{1})+m\frac{F_{\mathrm{mel}}(f_{h})-F_{\mathrm{mel}}(f_{l})}{M+1})$$
(37)$$F_{\mathrm{mel}}(f)=1125\log(1+f/700)$$

M=40 is the number of filters. For processing audio with 16 kHz sampling rate, $f_{h}=8000$, $f_{l}=0$, $f_{s}=16,000$ were used and $F_{\mathrm{mel}}^{-1}(f)$ is the inverse function of $F_{\mathrm{mel}}(f)$. Then, 161-dimensional features are compressed to 40-dimensions.

### 3.2. Learning Machine and Training Setup

Recurrent neural network (RNN) presents an outstanding performance in real-time speech-enhancement tasks [36]. Many recurrent architectures, including the long short-term memory (LSTM) [39] and the gated recurrent unit (GRU) [40] rely on gate to control the flow of sequence information to avoid the problem of exponential weight decay or explosion. Simple recurrent unit (SRU) has a better performance and higher computational efficiency than the GRU and LSTM, and it has strong parallel computing capabilities (in both forward propagation and back propagation) [41].

We stacked three layers of SRU and a fully connected layer with a sigmoid activation function as the output to estimate the band IRM as shown in Figure 6. The training label of the band IRM is:(38)$$g_{\mathrm{birm}}(b)=\sqrt{\frac{E_{s}(b)}{E_{x}(b)}}$$
where b is the band index, $E_{s}(b)$ and $E_{x}(b)$ are the clean and noisy band energies, respectively. As shown in Figure 6, we concatenated the features of the *t*-th frame, the (*t* − 1)-th frame, and the (*t* − 2)-th to the neural network, and estimated the band IRM gain of the *t* − 2th frame to better eliminate non-stationary noise. It is worth pointing out that this does not violate the real-time system’s rule of one frame in and one frame out, but it will cause a two-frame (20 ms) delay. The constant matrix (untrainable) interpolate matrix (IM) is deployed afterwards to interpolate the band IRM from 40 to 161 dimensions and the noisy signal in the complex frequency domain is multiplied with the estimated IRM, and finally the inverse fast Fourier transform (IFFT) is conducted to obtain the time-domain waveform estimation.

For the predicted band IRM, the mean squared error (MSE) is usually used as the loss function in the frequency domain, given by
(39)$$L_{1}={\left({\hat{g}}_{\mathrm{birm}}(b)-g_{\mathrm{birm}}(b)\right)}^{2}$$

In this work, we also propose to utilize the scale-invariant signal-to-distortion ratio (SI-SDR) [42] in time domain loss function to better minimize the distortion of model prediction, which: (40)$$L_{2}=-10\log_{10}(\frac{{||\frac{{\hat{s}}^{T}s}{{\left||s|\right|}^{2}}s||}^{2}}{{||\frac{{\hat{s}}^{T}s}{{\left||s|\right|}^{2}}s-\hat{s}||}^{2}})$$
where $\hat{s}$ represents the prediction of the time domain speech wave. 

Finally, to integrate two losses, the loss function of the network is weighted by $L_{1}$ and $L_{2}$, which can be written as
(41)$$L=\alpha L_{1}+(1-\alpha )L_{2}$$
where α is the factor that controls the weights of two losses. In practice, through training experiments, we found that the value α=0.3 provides a good trade-off between noise suppression and speech distortion.

## 4. Experimentation Details

### 4.1. System Processing Pipeline

The system pipeline is depicted in Figure 7, where the BC-assisted real-time speech-enhancement system proposed in this paper consists of two parts: the BC-assisted array signal processing and the SRU-based postfiltering. The former is used to suppress the directional point source interference, and the latter uses a computational efficiency neural network structure as a postfilter algorithm to further suppress residual noise. In Figure 7, $x_{0}(t)$, $x_{1}(t)$ and $x_{\mathrm{BC}}(t)$ are captured simultaneously. The processing of the sub-modules is carried out frame-by-frame synchronously, where STFT indicates short time Fourier transform and VAD means voice activity detection.

### 4.2. Performance Metrics 

We used two performance metrics, the perceptual evaluation of speech quality (PESQ) [43] and the short-time objective intelligibility (STOI) [44] to evaluate the speech quality and intelligibility, respectively. For these evaluation indicators, higher scores indicate the better results. The PESQ and STOI are all objective, and the evaluations require clean reference signals. Therefore, objective scores can only be performed on the synthetic test set. Research in [45] pointed out that there is still a gap between the objective scores and human’s subjective hearing, so we use data recorded in real scenes and subjective scores are also provided as a supplement to performance measures [45]. The listener is given a score of 1 (very poor sound quality) to 5 (very good sound quality), and each listener is required to go through scoring training and pass the qualification test with a scored (labeled) dataset before scoring. We averaged the scores to obtain the mean opinion score (MOS).

### 4.3. Comparisons

The method proposed in this paper integrates an array-based adaptive beamformer and a DNN-based postfilter. In order to fully verify the superiority of the proposed algorithm and show the effectiveness of the BC signal assistance in the algorithm performance, we split the proposed algorithm into the BC–GSC module: BC-assisted array signal processing, and the SRU-P module: SRU-based postfiltering to compare with some relevant algorithms. The whole system is named BC–SE. In details, we compare proposed BC–GSC module with two baselines, i.e., the robust (R)-GSC [33], and the transfer-function (TF)–GSC [46]. For the evaluations of the module SRU-P, we selected OMLSA [5] as the conventional method baseline, and RNNoise [7] as the deep learning method baseline. OMLSA is a speech-enhancement algorithm based on digital signal processing (DSP) which is widely used in engineering because of its robustness and low computational complexity. RNNoise is the state-of-the-art deep learning-based single-channel real-time algorithm with a number of parameters less than 100 K. For comparison with whole BC–SE system, the postfilter transient beam-to-reference ratio (TBRR) [27] and RNNoise are combined with basic GSC and R-GSC to form complete array processing postfilter noise reduction system baselines.

### 4.4. Experimental Setting

In the experiments, the LIS25BA from STMicroelectronics was selected as the BC sensor which is a high-bandwidth three-axis digital accelerometer chip. The price of LIS25BA is about 2 USD. The LIS25BA is available in a small thin plastic land grid array (LGA) package. The development board with dual-microphone and LIS25BA can be seen in Figure 8. LIS25BA is attached on the ear back side to collect the BC signals. The microphone array end-fire direction, along the axis of the array is pointed to the speaker’s mouth with a distance of 10 cm to record clean speech in a reverb-free environment. The STM32 development board is employed to synchronize the capture of the BC and AC signals. In order to objectively assess the performance of the proposed algorithm, the noise and clean speech were first separately recorded and then mixed at different SNRs (−5 dB, 0 dB, 5 dB, 15 dB) with room reverberation. The babble noise generated by a number of talking speakers and car cabin noise are recorded as the diffuse non-directional interference. The music noise and speech-shaped noise are recorded as the directional interferences (60°, 90°, 180°). The microphone array end-fire direction is pointed to the speaker’s mouth with a distance of 10 cm to record the clean speech in a reverb-free environment. LIS25BA is attached to the ear back side to collect the BC signals. We recorded 20 clips of each type of noise, and the average length of each clip was 5 s, which was used to synthesize the noisy data. In addition, we recorded 15 clips of each type of noise in real scenes for subjective evaluation testing. We used the deep noise suppression challenge (DNS)dataset [47] to train the proposed SRU-based network and RNNoise network, and used the Adam [48] optimizer to train with a learning rate of 0.001 and a batch size of 128 and 10 epochs. The original RNNoise was also modified to process a 16 kHz sampling rate audio. We used DNS data collection to form a 500 h training set (single channel, 16 kHz sampling rate, SNR ranging from −5 dB to 30 dB). The DNS test set (synthetic) was used to evaluate the performance of the SRU-based postfilter. In this work, the frame size was 20 ms and the shift was 10 ms (50% overlap), and the FFT size was equal to the frame size.

## 5. Results and Discussions 

### 5.1. Performance Evaluation of BC–GSC

Thanks to the wind noise suppression module based on the BC signal, the wind noise suppression performance is excellent, which can be determined from the spectrogram in Figure 9 The GSC performance for the AC-only signal is good against point source interference and can significantly improve the voice quality, but the performance for non-point source noise is limited. GSC assisted by BC signal effectively enhances the suppression of non-point source noise at a low SNR.

A clean female speech was mixed with SNR = 0 dB music noise. Three enhancement approaches were applied to obtain results as shown by the spectrograms in Figure 10. It can be observed the R-GSC effectively suppressed the directional noise above 2 kHz, and the BC–GSC better suppressed the directional noise between 1 and 2 kHz. The noise in the non-existent segment of the speech was effectively eliminated. The evaluation metric for each algorithm is obtained on the test set described above, shown by the histograms in Figure 11. For non-point source noise, the performance of AC-only GSC is unsatisfactory. The BC–GSC in the case of car noise and wind noise is significantly better than that of the R-GSC, and TF–GSC. This is because in a diffuse noise scenario, the latter two algorithms have difficulty distinguishing the interference and the desired signal, resulting in the failure of an adaptive filter update. Due to the BC signal that is not affected by sound field noise, the advantage of the proposed approach at low SNR is more significant.

### 5.2. Performance Evaluation of Postfiltering

Table 1 shows the evaluation performance of OMLSA, RNNoise and our proposed method and the best performing scores are marked in boldface. It can be seen that the DNN-based method is significantly better than the traditional OMLSA method, because the conventional method depends on the assumption that noise changes much more slowly than speech. However, the DNN-based methods do not require this assumption. The proposed method is slightly better than RNNoise at a high SNR, but significantly better than it at a low SNR. This is due to two reasons. One is that the proposed sub-band density is nearly twice that of RNNoise (Proposed: 40, RNNoise: 22), so the spectral resolution is higher. The second is that the usage of two additional frames, as a cost of 20 ms delay, increases the ability to estimate noise and protects the speech component. Since the number of parameters for the proposed method and for RNNoise is almost the same, both can be processed in real time by low-power devices.

### 5.3. Performance Evaluation of the Proposed Speech Enhancement Algorithm

We averaged the PESQ, STOI and MOS of various noises in the test set. It can be seen from Table 2 that the proposed system is superior to the baseline in terms of subjective and objective evaluation. This is consistent with the results for the separate modules as shown in the previous sections of the performance evaluations. It demonstrates that the proposed speech enhancement algorithm is effective.

## 6. Conclusions

This paper proposes a new algorithm using BC signals to assist dual-microphone GSC adaptive beamforming for speech enhancement. First, the BC signals were used to conduct highly reliable VAD-assisted ANC and ABM weight coefficient updates in GSC. Second, an adaptive filter CF is designed to compensate the amplitude and phase difference between AC and BC signals. Third, wind noise is detected and replaced with the output of CF to recover low-frequency speech components from the wind noise. Finally, a real-time neural network-based postfilter is designed and trained to effectively remove the residual noise. Experimental results show that the proposed algorithm effectively improves SNR and the speech quality in different scenarios, and the assistance of BC signals can effectively improve the noise reduction performance of beamforming. 

## Figures and Tables

**Figure 1 sensors-20-05050-f001:**
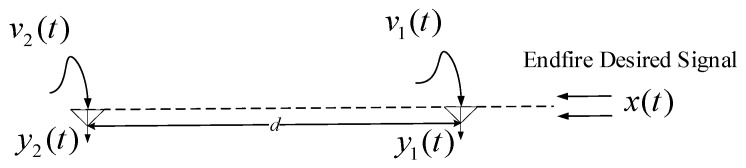
Dual-microphone array with end fire desired signal.

**Figure 2 sensors-20-05050-f002:**
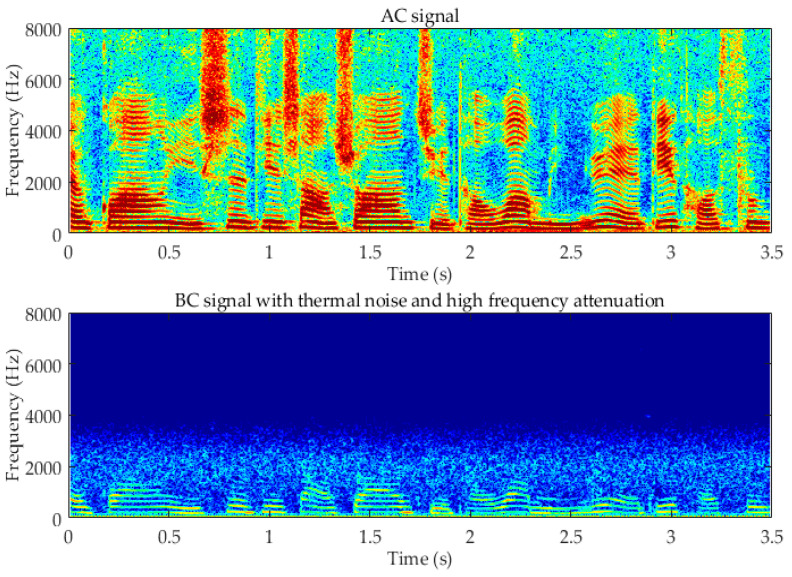
Clean air-conduction (AC) signal and bone-conduction (BC) signal.

**Figure 3 sensors-20-05050-f003:**
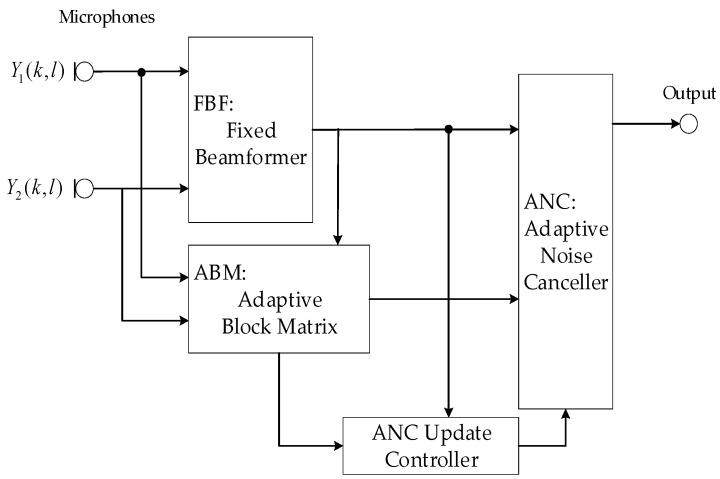
The robust generalized sidelobe canceller (GSC) with coefficients constrained ABM structure.

**Figure 4 sensors-20-05050-f004:**
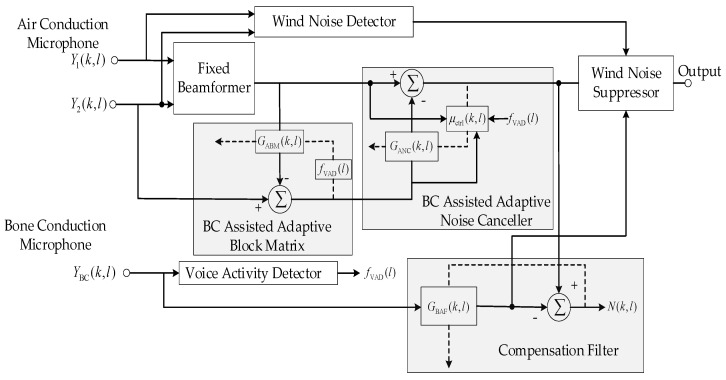
The framework of the proposed BC signal-assisted adaptive beamforming algorithm for speech enhancement.

**Figure 5 sensors-20-05050-f005:**
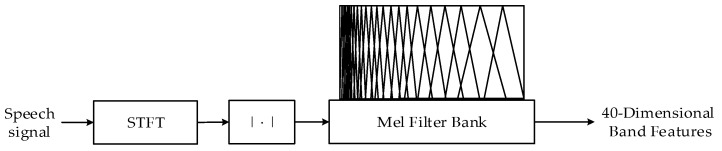
The scheme for extracting Mel-scale sub-band features.

**Figure 6 sensors-20-05050-f006:**
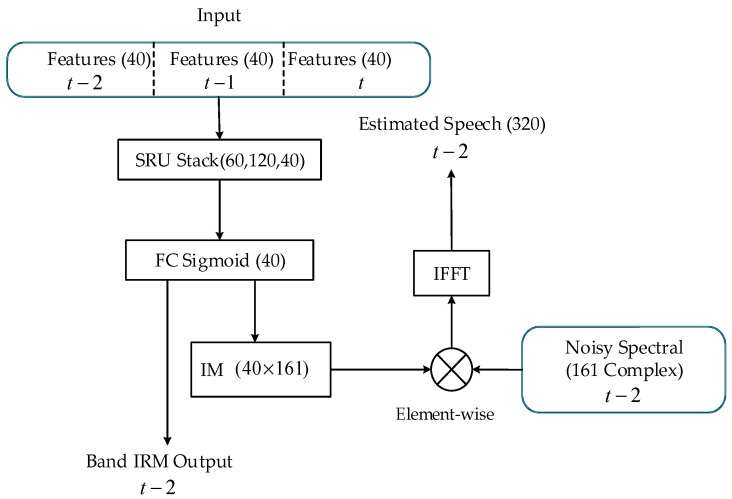
The diagram of the proposed Mel-scale sub-band simple recurrent unit (SRU)-based neural network. FC means the fully connection. IM indicates interpolate matrix. IFFT means Inverse fast Fourier transform and IRM is ideal ratio mask.

**Figure 7 sensors-20-05050-f007:**
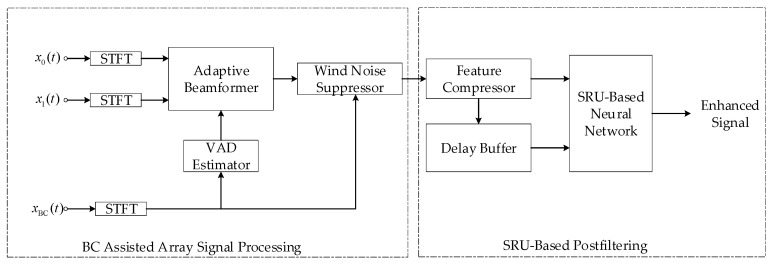
The system pipeline is divided into two parts: BC-assisted array signal processing and SRU-based postfiltering.

**Figure 8 sensors-20-05050-f008:**
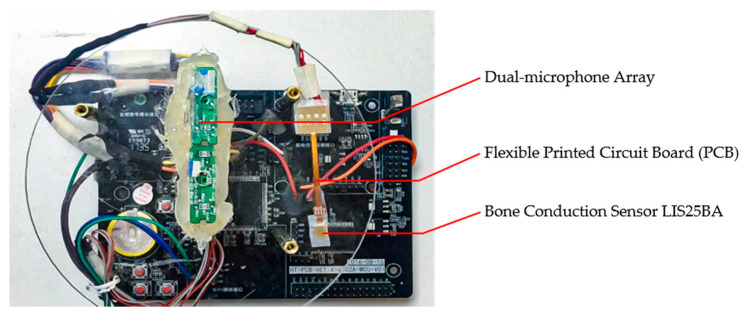
The STM32 development board with the dual-microphone array and the bone conduction sensor LIS25BA.

**Figure 9 sensors-20-05050-f009:**
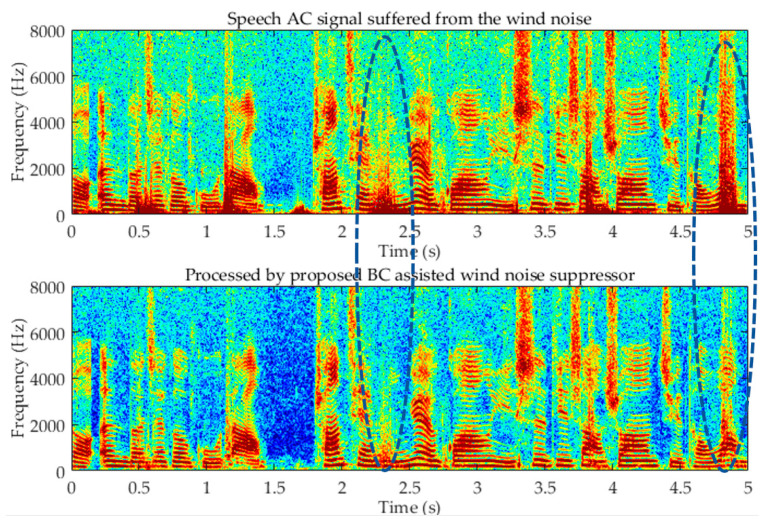
Wind noise processed by the proposed dual-microphone BC sensor-assisted method.

**Figure 10 sensors-20-05050-f010:**
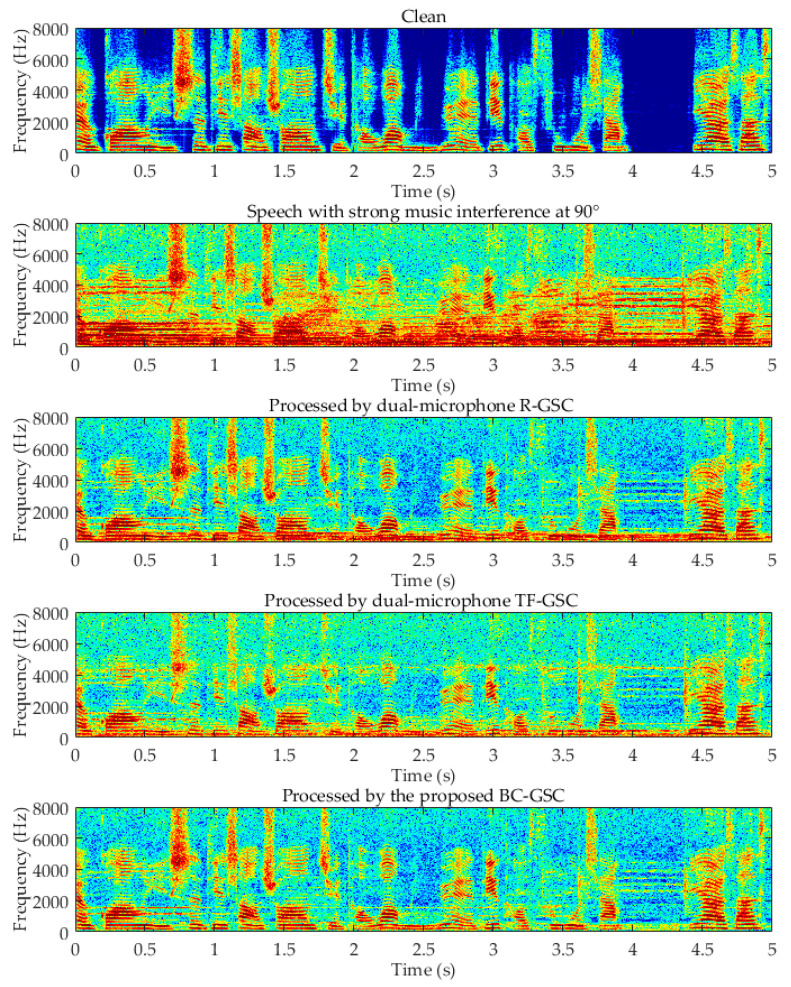
Spectrograms of an utterance corrupted by 0 dB music interference at 90° and the processing results by the robust (R)-GSC, the transfer-function (TF)–GSC and the proposed BC–GSC methods, respectively.

**Figure 11 sensors-20-05050-f011:**
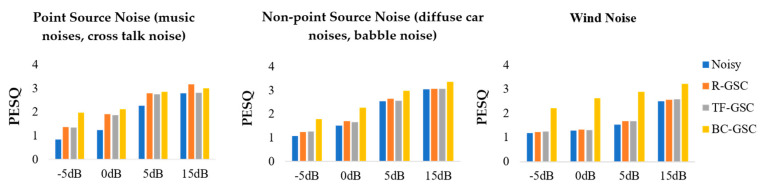
The perceptual evaluation of speech quality (PESQ) of the R-GSC, the TF–GSC and the proposed BC–GSC methods in different noises.

**Table 1 sensors-20-05050-t001:** The postfilter (single-channel) speech enhancement results obtained with optimally modified log-spectral amplitude (OMLSA), Recurrent neural network (RNN) noise and the proposed method, where noisy means the unprocessed original signal.

	SNR 0–5 dB				SNR 5–20 dB			
	Noisy	OMLSA	RNNoise	SRU-P	Noisy	OMLSA	RNNoise	SRU-P
PESQ	1.75	1.77	2.43	**2.51**	2.40	2.64	2.86	**2.94**
STOI (%)	76.5	76.0	76.7	**77.9**	88.2	88.3	88.5	**88.7**
Parameters	None	Oracle	88 k	89 k	None	Oracle	88 k	89 k

**Table 2 sensors-20-05050-t002:** The objective and subjective measurements comparing the PESQ and MOS performance among TF–GSC+TBRR, TF–GSC + RNNoise, R-GSC + RNNoise and BC–SE.

**0–5 dB**					
	**Noisy**	**TF–GSC + TBRR**	**TF–GSC+RNNoise**	**R-GSC+RNNoise**	**BC–SE**
PESQ	1.18	1.29	1.52	1.55	**1.64**
STOI (%)	71.3	71.4	70.6	70	**74.3**
MOS	2.54	2.71	3.56	3.45	**3.64**
**5–15 dB**					
	**Noisy**	**TF–GSC+TBRR**	**TF–GSC+RNNoise**	**R-GSC+RNNoise**	**BC–SE**
PESQ	1.87	1.95	2.15	2.19	**2.24**
STOI (%)	76.4	76.9	76.1	76.9	**77.1**
MOS	3.75	3.84	**3.87**	3.82	3.86

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
