# Peer review of "A Real-Time Dual-Microphone Speech Enhancement Algorithm Assisted by Bone Conduction Sensor"

_sensors, 2020, doi:10.3390/s20185050_

Round 1

Reviewer 1 Report

The manuscript presents a real-time speech enhancement system. The paper is well-written and include current and relevant references. The authors describe in detail the proposed system and the experiments done. The results show the effectiveness of the proposed technique.

I propose to the authors some ideas to increase the readability of the paper, to avoid some confusing points and to highlight the novelty and effectiveness of the proposed work:

1.- ¿How is the value of alpha in Eq. (39) determined? I think that this point should be clarified.

2.- In my opinion, the pipeline system shown in Fig.6 requires an explanation. In a conventional pipeline, the task is divided into different subtacks so that each stage of the pipeline performs a different subtask related to data from different instants of time (t0, t1, t2, ...). This allow to execute in parallel the different subtasks, increasing the throughput. I think that it is not clear how the proposed system is pipelined.

3.- For clarity, I think that it would be a good idea to include the definition of MOS in Section 4.2 instead of in the caption of Table 2.

4.- I believe that the authors should define "Noisy" in Table 1 and 2. It can be confusing because "Noisy" represents the original signal and not a method.

5.- The author should explain clearly the different between the proposed technique and the state-of-the-art methods. In this sense, I think that it is important that the authors justify why the comparison have been done with RNNoise [7] and OMLSA[5]; do they represent the state-or-the-art of this kind of systems. In my opinion, this point should be clarified.

Author Response

Please see the attched response letter. Thanks.

Reviewer 2 Report

  A more Deep explenation on BC sensor setting will be useful for sake of clarity

Author Response

(The authors gave the same response as above.)

Reviewer 3 Report

The manuscript presents an interesting method to use bone-conducted signals to reduce background noise in acoustic signals collected by wearable smart devices. Presumably, a goal is to reduce the effect of wind noise and vehicle compartment noise. One wonders whether the method would have value also for persons with hearing loss. The reviewer has the following suggestions/concerns that the authors may wish to address:

Line 45 “effective to suppress”

Line 61 “distortionless” should be “distortionfree”

Line 66 “a signal-blocking matrix (BM)

Line 75 “is relatively insensitive”

Line 98 “the array postfiltering algorithm has difficulty obtaining benefit”

Line 103 “An improved VAD is extracted from the BC”

Line 128 “fixed in the endfire direction, along the axis of the array”

Line 214 “reasonably compromise noise suppression and voice quality” probably should be “provide a reasonable compromise between noise suppression and voice quality”

Line 322 please explain further what is babble noise.

Line 346 “is well for point” probably should be “is good against point”

Line 357 “the latter two algorithms have difficulty distinguishing the “

Line 372 At what level of ms delay do listeners start noticing. Many people can pick up 20 ms delays, but perhaps the delays are not so problematic for noise canceling operations.

Line 375 by low-power devices.

Author Response

(The authors gave the same response as above.)
